# MSGWO-MKL-SVM: A Missing Link Prediction Method for UAV Swarm Network Based on Time Series

**Mingyu Nan** , **Yifan Zhu, Jie Zhang, Tao Wang and Xin Zhou** *

College of System Engineering, National University of Defense Technology, Changsha 410000, China; nanmingyu@nudt.edu.cn (M.N.); yfzhu@nudt.edu.cn (Y.Z.); zhangjie@nudt.edu.cn (J.Z.); wangtao1976@nudt.edu.cn (T.W.)
* Correspondence: zhouxin09@nudt.edu.cn

**Abstract:** Missing link prediction technology (MLP) is always a hot research area in the field of complex networks, and it has been extensively utilized in UAV swarm network reconstruction recently. UAV swarm is an artificial network with strong randomness, in the face of which prediction methods based on network similarity often perform poorly. To solve those problems, this paper proposes a Multi Kernel Learning algorithm with a multi-strategy grey wolf optimizer based on time series (MSGWO-MKL-SVM). The Multiple Kernel Learning (MKL) method is adopted in this algorithm to extract the advanced features of time series, and the Support Vector Machine (SVM) algorithm is used to determine the hyperplane of threshold value in nonlinear high dimensional space. Besides that, we propose a new measurable indicator of Multiple Kernel Learning based on cluster, transforming a Multiple Kernel Learning problem into a multi-objective optimization problem. Some adaptive neighborhood strategies are used to enhance the global searching ability of grey wolf optimizer algorithm (GWO). Comparison experiments were conducted on the standard UCI datasets and the professional UAV swarm datasets. The classification accuracy of MSGWO-MKL-SVM on UCI datasets is improved by 6.2% on average, and the link prediction accuracy of MSGWO-MKL-SVM on professional UAV swarm datasets is improved by 25.9% on average.

**Keywords:** UAV swarm; missing links prediction; time series data; multiple kernel learning; multi-objective optimization; grey wolf optimizer; support vector machine; complex network

**MSC:** 68T20

## 1. Introduction

In modern intelligent warfare, the saturation assault of a UAV swarm is one of the most severe threats to the defender in many typical defense scenarios. It has been very urgent to develop an effective countermeasure to defend the saturation assaults of UAV swarm [1]. One purpose of link prediction in an UAV swarm network is to reconstruct its communication topology structure, so that a UAV swarm network disintegration strategy can be generated later. Recent studies have borne out that the fire control scheme based on the network disintegration strategy will improve the defensive effectiveness of UAV swarm interception significantly. The UAV swarm network disintegration strategy is shown in Figure 1.

A UAV swarm consists of a group of isomorphic or heterogeneous UAVs. The current UAV swarm command and control system still adopts the man-in-the-loop mode. The UAV swarm system receives the task instructions from the commander in the OODA (Observe, Orient, Decide, and Act) loop and completes the task semi-autonomously. The UAV swarm system is unable to be out of the commander's control completely.

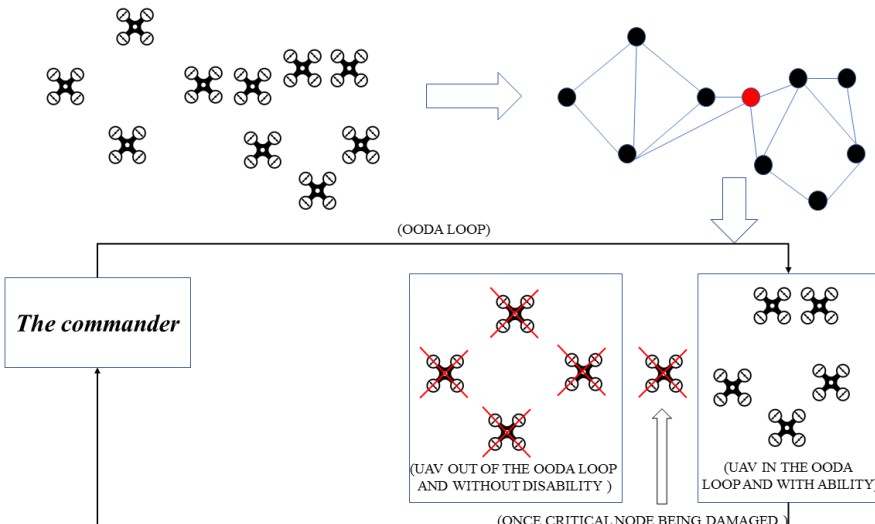

**Figure 1.** The flow of UAV swarm network disintegration strategy.

For the commander, the connectivity of the UAV swarm network is a prerequisite to realizing its characteristics of swarm intelligence and task coordination, and the two-connectivity character is a basic requirement for UAV swarm networks in a battlefield environment. In a UAV swarm, some nodes play an important role in the network's two-connectivity characteristics, which are called Critical Nodes, and the cut-vertex is one of those Critical Nodes.

The red node in Figure 1 is a cut-vertex of the given UAV swarm. Once the red node is damaged, the UAV swarm will be quickly split into multiple subblocks, and a large number of UAVs will break away from the network, not able to continue receiving the task instructions of a commander. At that time, the combat effectiveness of the UAV swarm will be greatly reduced.

UAV swarm network communication topology reconstruction is one of the technical routes to realize the identification of Critical Nodes of the UAV swarm network. The main contribution of this paper concentrates on reconstructing the UAV swarm network topology structure by the method of predicting missing links in complex networks. Missing link prediction (MLP) is a microscopic prediction in a complex network. Instead of predicting the global properties such as modularity, degree, and clustering coefficient, its goal is to assess the existence probability of each non-observed link, according to the known information from network structure and individuals' attributes [2,3].

So far, scholars have proposed many effective missing link prediction methods [4,5], in which the similarity-based methods are utilized widely, such as the Common Neighbors algorithm [6–8], the Adamic-Adar index algorithm [9–11], and the Local Random Walk algorithm [12]. Newman M. E. J. [6] studies empirically the time evolution of scientific collaboration networks in physics and biology. It shows that the probability of a pair of scientists collaborating increases with the number of other collaborators they have in common. Lada A. Adamic [9] attempts to predict the links among the activities of the internet network and reflect those activities of the internet network into the real world. He proposes some effective indicators of social connections and finds that these indicators vary drastically in different user populations. Those indicators generate far-reaching influence and have great significance in the field of MLP technology. However, as complex networks increase in size and node numbers, the computing burden of MLP technology has got heavier and heavier. To overcome the difficulties of the sparsity and huge size of the target networks, Liu [12] proposes a local random walk strategy, which can give better prediction while having a much lower computational complexity. Similarity-based MLP technology has also been utilized in many other networks, such as counter-terrorism [13], e-commerce [14], biological [15,16], and social [17–19].

However, similarity-based methods are mainly applicable to complex networks with some regularities, such as scale-free networks [20], regular networks, and small-world networks [21]. As for the UAV swarm system, it is an artificial network with strong randomness and high uncertainty, and the similarity-based method is hard to predict the missing link accurately for this scenario.

Current studies on complex network link prediction mainly focus on static and deterministic networks. In fact, many networks in the real world are dynamic and time-varying. For example, the UAV swarm network proposed in this work will evolve with the relative position of the UAV. The disintegration strategy and link prediction of time-varying complex networks is a challenge in the future [22]. Ren assumed that the UAV swarm would take some time to form a flying formation before performing reconnaissance, attack, and other tasks. During this period, the data link of the UAV swarm network can be predicted. However, Ren did not give a specific link prediction method and directly gave the disintegration strategy of the UAV swarm network under the premise of the known network links and topology reconstruction [23]. Shu and Qi conducted in-depth research on the problems of UAV swarm network link prediction independently. Aiming at the characteristics of the UAV swarm network, Shu proposed a temporal graph embedding model to reconstruct the UAV swarm network. The network structure features were mapped to the relationships between nodes, and the contextual semantic features of nodes were extracted by adversarial training. With the help of long and short-term memory networks, the temporal characteristics of the UAV swarm network are extracted to predict the link [24]. However, the method employed by Shu, is a black-box model that lacks interpretability and does not take full advantage of the velocity information of the UAV swarm network. Qi proposed a Markov chain-based link prediction algorithm for the UAV swarm network topology, which could predict the link between a pair of nodes. For the convenience of analysis, Qi employed a SYN-boid model to describe the swarming motions of nodes in the UAV swarm network [25]. The most important transition probability matrix is hard to obtain in the method of Qi. In order to make up for the defects of the method proposed by Shu and Qi, an equivalent transformation model from time series to complex network was adopted in this paper.

For a complex system, there are two paradigms to describe its intrinsic dynamic behavior, namely time series and complex networks. Hence, this paper attempts to develop a new missing link prediction technology, applicable to UAV swarm networks, based on the time series. In fact, the method of transformation from time series to complex networks has been developed for decades and could be summarized in the following three main categories: Neighbor-Joining [26,27], Visibility Graph [28–30], and Transfer [31,32].

Zhang and Small [26] first proposed converting time series into complex networks for analysis in 2006. They construct complex networks from pseudo-periodic time series, with each cycle represented by a single node in the network. Two nodes are deemed to be connected if the phase space distance between the corresponding cycles is less than a predetermined value *D*. On the basis of Zhang and Small, Yang [28] proposes a method of fixed length segmentation of time series and examines the correlation coefficient of individual time series segments. In 2014, Zhao [33] proposes a magnitude difference mapping method from one-dimensional time series to complex networks. The magnitude difference mapping method directly examines the magnitude difference between time series with respect to a given threshold value, so it provides the possibility of proving the equivalence relation between time series and complex networks, analytically. Peng [34], in 2020, studied the transformation between time series and complex networks both in theory and applications. He analyzes the equivalence of the amplitude difference mapping method from time series to complex networks. Two quasi-isometric isomorphism theories of metric space are summarized and their relations are found in the literature [35]. However, the analytical proof work is only based on a series of ideal non-linear differential equations, with the equation parameters and variables not having definite physical meanings. The overidealized mathematical model is a little far away from the real situation in the real world.

The preceding works have made significant contributions to the MLP problems of UAV swarm networks based on time series, but they also have the following issues:

1. The operations of the above algorithms of transformation between time series and complex networks are all processed in ordinary Euclidean space. However, many advanced relational features of the UAV swarm network's dynamic behaviors may not be obvious in ordinary Euclidean space;
2. The dimension of extracted features about time series is relatively low in the above algorithms of transformation between time series and complex networks, which may not be able to characterize the UAV swarm network sufficiently;
3. The determination of thresholds relies heavily on the experience of researchers and is absent with efficient methods, as used in the above algorithms of transformation between time series and complex networks.

To effectively solve the aforementioned problems, a multi-kernel learning algorithm based on a multi-strategy grey wolf optimizer (MSGWO-MKL-SVM) was proposed, and the framework of MSGWO-MKL-SVM is shown in Figure 2.

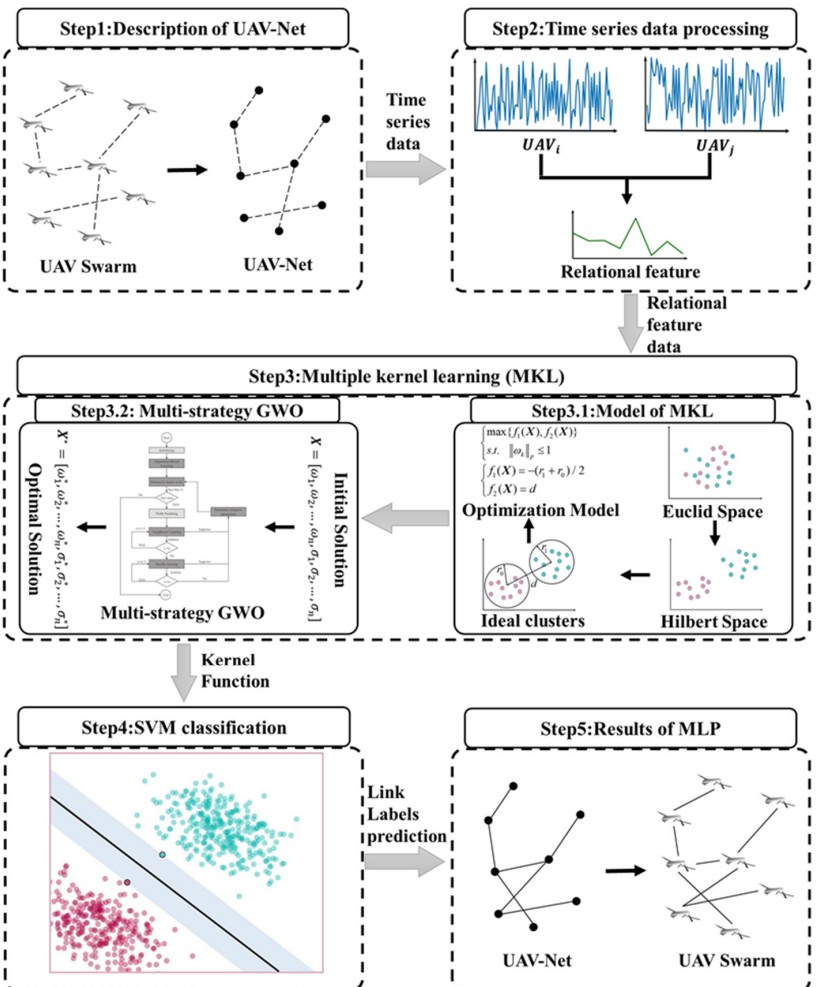

**Figure 2.** Framework of the MSGWO-MKL algorithm.

The MSGWO-MKL-SVM algorithm for predicting links of UAV swarm networks based on time series contains five steps. In step 1, the relationship between UAV swarms and complex networks was described. Nodes of networks represent UAVs, and edges of networks represent the communication links among a UAV swarm.

In step 2, the time series data of a UAV swarm were processed. The advanced relational features of UAV-*i* and UAV-*j* from the training set were extracted by correlation analysis.

In step 3, new indicators of multi-kernel learning were established based on the idea of clusters, transforming a multi-kernel learning problem into a multi-objective optimization problem. Besides that, a multi-strategy grey wolf optimizer algorithm was proposed to solve this problem, whose global searching ability was greatly enhanced compared with some state-of-the-art algorithms. In step 4, the trained kernel functions were put into the Support Vector Machine (SVM) algorithm, extracting the advanced features of time series in high dimensional space, and then determining the threshold hyperplane. In step 5, the time series data of two UAV-*i* and UAV-*j* from the test set were fed into the MSGWO-MKL-SVM algorithm, which determined whether UAV-*i* and UAV-*j* had a communication link.

The main contributions of this paper can be summarized as follows:

1.  The multiple kernel learning (MKL) method was adopted to transform the features of time series from a linear Euclidean space to a non-linear Hilbert space, making the features of time series of a UAV swarm network more remarkable in the kernel space;
2.  New indicators of multi-kernel learning (MKL) were established based on the idea of clusters, transforming a multi-kernel learning problem into a multi-objective optimization problem and reducing the computational complexity greatly;
3.  Variable neighborhood search strategies and parameters adaptive operators were designed to enhance the global searching ability of grey wolf optimizer algorithm (GWO). Besides that, we adopted the opposition-based learning strategy to increase the diversity of the initial population. Eventually, a multi-strategy grey wolf optimization algorithm (MSGWO) was proposed in this paper. The MSGWO algorithm can enhance the balance between local and global search and maintain diversity. In the standard UCI dataset, the MSGWO algorithm performed better than some state-of-the-art algorithms;
4.  Multiple kernel learning and Support Vector Machine (MKL-SVM) algorithms were adopted to calculate the threshold hyperplane of correlation features of the UAV swarm network in the kernel space directly, avoiding empirical estimation of the threshold;
5.  The multifractal detrended cross-correlation analysis (MF-DCCA) method was used to analyze the correlation of time series between UAV-*i* and UAV-*j*. *K*-order cross-correlativity coefficient was defined to extract high-dimensional features of cross correlativity between UAV-*i* and UAV-*j*.

The rest of this paper is organized as follows. In Section 2, basic descriptions of the UAV swarm network and the missing link prediction (MLP) model are established. In Section 3, the time series data process and cross-correlation analysis are described. In Section 4, new indicators of multi-kernel learning (MKL) were established based on the idea of clusters, and a multi-strategy grey wolf optimization algorithm (MSGWO) is proposed. Sections 6 and 7, calculating samples are provided and discussed. In Section 8, several conclusions are given, in addition to a discussion on future research.

## 2. Description of UAV Swarm Networks

There is a UAV network containing *n* UAVs and the control equation of the UAV-*i* is represented in Equation (1) [36]:

$$\dot{\xi}_i = \varsigma_i, i = 1, \dots n \qquad (1)$$

where $\xi_i$ represents the displacement statement of the UAV-*i*; $\varsigma_i$ represents the velocity statement of the UAV-*i*.

A graph $G = (V, E)$ describes a directed communication network structure of the UAV swarms. The graph $G$ consists of a node set $V(G)$ and an edge set $E(G)$, where an edge is an ordered pair of distinct nodes in the graph $G$. One node of the network represents one UAV in the swarm, and one edge between two distinct nodes represents that there exists a message delivery between the two UAVs.

It is assumed that the matrix $A(G) = [a_{ij}]$ is a $n \times n$ matrix, which is termed the adjacency matrix of the UAV network, and follows Equation (2) [37].

$$a_{ij} = \begin{cases} 1, \ if \ e_{ij} \in E(G) \\ 0, \ otherwise \end{cases} \tag{2}$$

where $a_{ij}$ represents the message delivering conditions from the UAV-$i$ to the UAV-$j$; $e_{ij}$ is an edge between node-$i$ and node-$j$.

In order to simulate the behavior of bird swarms, Reynolds proposed the Boid model in 1986 [38], in which swarms meet the following three principles:

1. Separation: One node is too close to another node in its repulsion zone, and the two nodes will repulse and move in opposite directions;
2. Aggregation: Each node will move closer to the central node of the swarm;
3. Coherence: Each node will adjust its speed and direction of movement, based on its neighbor nodes, to ensure the velocity consistency of the whole swarm.

Based on the above three principles, the formation and topology control illustration of UAV swarms are as in Figure 3.

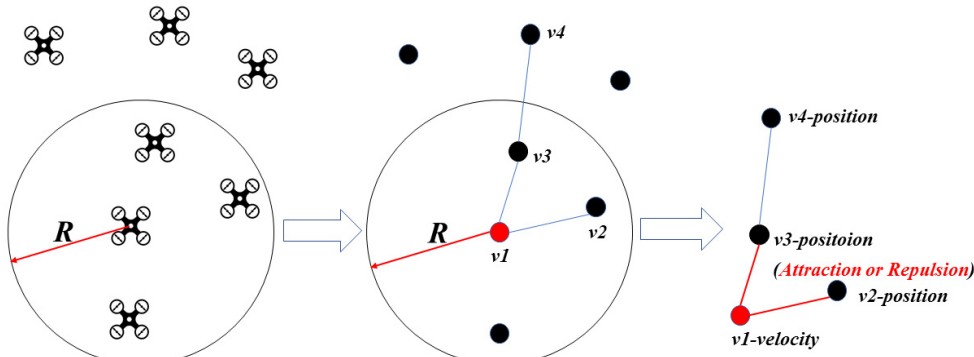

**Figure 3.** Formation and topology control illustration of UAV swarms.

The UAV node could establish data links with other nodes in its communication radius, R. However, limited by the number of channels and communication quality, each UAV node can only establish data links with part of the nodes. In Figure 3, The UAV node $v1$ only establishes connections with $v2$ and $v3$. Based on Boid's three principles, the relative distance and direction between nodes $v3$ and $v1$ will affect the velocity vector of node $v1$. Although node v4 can also exert indirect influence on node $v1$ through node $v3$, the influence degree is weaker than the direct influence of node $v3$. For node $v1$, the indirect influence degree of K-hop node v4 and the direct influence degree of 1-hop node $v3$ is different. The following work in this paper mainly focuses on the quantification of the influence degree between nodes and the determination of the threshold of direct influence of 1-hop nodes and indirect influence of K-hop nodes.

Therefore, $\{\xi_i(t), \xi_j(t)\}_{i=1}^n$ is defined as a set of time series of the UAV swarm network, $\xi_i(v_i, t)$ is the velocity time series of the UAV-$i$, $\xi_j(x_{ij}, t)$ is the relative displacement time series of the UAV-$i$ and UAV-$j$, and $\varepsilon$ is a threshold. We use the absolute metric of time series in high dimensional nonlinear Hilbert space as a transformation method, as in Equation (3).

$$a_{ij} = \begin{cases} 1, \ if \ \| \phi[\xi_i(t), \xi_j(t)] \| \geq \varepsilon \\ 0, \ otherwise \end{cases} \tag{3}$$

where $t$ is the time, $\| \cdot \|$ is an absolute metric function, and $\phi(\cdot)$ is a mapping function.

### 3. Time Series Data Processing and Cross-Correlation Analysis

A Detrended Cross-Correlation Analysis (DCCA) algorithm [39] is proposed by Podobnik. The DCCA algorithm is designed to measure the cross correlativity about two nonstationary time series. The Multifractal Detrended Cross-Correlation Analysis (MF-DCCA) method [40,41] integrates multifractal theory on the basis of DCCA, measuring the multifractal properties about two nonstationary time series in different timescales. In this section, a new high-order cross correlativity coefficient based on MF-DCCA was proposed, which aims to extract the high dimensional cross correlativity features of the two nonstationary time series of UAV-*i* and UAV-*j*.

A construction equation for two new time series, $\{\check{\xi}_i(t)\}_{t=1}^T$ and $\{\check{\xi}_j(t)\}_{t=1}^T$, based on the UAV swarm network's dynamic behaviors, $\{\xi_i(v_i,t)\}_{t=1}^T$ and $\{\xi_j(x_{ij},t)\}_{t=1}^T$, about UAV-*i* and UAV-*j*, was devised. The parameter $T$ is the length of the time series. The construction equation is represented in Equation (4).

$$\begin{cases} \check{\xi}_i(t) = \sum_{k=1}^t \left( \xi_i(k) - \overline{\overline{\xi_i}} \right) \\ \check{\xi}_j(t) = \sum_{k=1}^t \left( \xi_j(k) - \overline{\overline{\xi_j}} \right) \end{cases} \tag{4}$$

where, $\overline{\overline{\xi_i}}$ and $\overline{\overline{\xi_j}}$, are the mean value of $\{\xi_i(t)\}_{t=1}^T$ and $\{\xi_j(t)\}_{t=1}^T$.

The two new time series, $\{\check{\xi}_i(t)\}_{t=1}^T$ and $\{\check{\xi}_j(t)\}_{t=1}^T$, were equally divided into $T_s$ inter-cells, as in Equation (5).

$$T_s = T/s \tag{5}$$

where, $s$, is the separation scale and $T_s$ is the separation number.

We defined inter-cell time series $\check{\xi}_i[t,m]$ and $\check{\xi}_i[t,m]$, in Equation (6).

$$\begin{cases} \check{\xi}_i[t,m] = \left[ \check{\xi}_i(t), t = 1, \dots, T_s \right]_m \\ \check{\xi}_j[t,m] = \left[ \check{\xi}_j(t), t = 1, \dots, T_s \right]_m \end{cases} \tag{6}$$

where parameter, $t$, is from 1 to $T_s$ and parameter $m$ is from 1 to $s$.

We defined the detrended inter-cell time series $\hat{\xi}_i[t,m]$ and $\hat{\xi}_j[t,m]$, in Equation (7).

$$\begin{cases} \hat{\xi}_i[t,m] = \check{\xi}_i[t,m] - \xi_i^v[t,m] \\ \hat{\xi}_j[t,m] = \check{\xi}_j[t,m] - \xi_j^v[t,m] \end{cases} \tag{7}$$

where, $\xi_i^v[t,m]$, is the trend function of the time series $\check{\xi}_i[t,m]$ and $\xi_j^v[t,m]$ is the trend function of the time series $\check{\xi}_j[t,m]$.

The *k*-order co-variance detrend fluctuant function and *k*-order variance detrend fluctuant functions about the two time series, UAV-*i* and UAV-*j*, follow Equation (8).

$$\begin{cases} F_{k-YY}(s) = \left\{ \frac{1}{s} \sum_{m=1}^s \left[ \frac{1}{T_s} \sum_{t=1}^{T_s} \left[ \xi_j[t,m] \times \xi_j[t,m] \right] \right]^k \right\}^{\frac{1}{k}} \\ F_{k-XX}(s) = \left\{ \frac{1}{s} \sum_{m=1}^s \left[ \frac{1}{T_s} \sum_{t=1}^{T_s} \left[ \hat{\xi}_i[t,m] \times \xi_i[t,m] \right] \right]^k \right\}^{\frac{1}{k}} \\ F_{k-XY}(s) = \left\{ \frac{1}{s} \sum_{m=1}^s \left[ \frac{1}{T_s} \sum_{t=1}^{T_s} \left[ \xi_i[t,m] \times \xi_j[t,m] \right] \right]^k \right\}^{\frac{1}{k}} \end{cases} \tag{8}$$

where, $F_{k-YY}(s)$, is the *k*-order of the covariance detrend fluctuant function, and $F_{k-YY}(s)$ and $F_{k-XX}(s)$ are the *k*-order of the variance detrend fluctuant functions.

The *k*-order cross correlativity coefficient was defined in Equation (9).

$$\rho_k = F_{k-XY}(s)^2 / F_{k-XX}(s) \times F_{k-YY}(s) \tag{9}$$

where $\rho_k$ is the *k*-order cross correlativity coefficient.

Eventually, the two time series about UAV-*i* and UAV-*j* were transformed into a $1 \times k$ dimension matrix $[\rho_{ij}]_{1 \times k}$, in Equation (10).

$$\left[\{\xi_i(t)\}_{t=1}^T; \{\xi_j(t)\}_{t=1}^T\right] \to [\rho_{ij}]_{1 \times k} = \left[\rho_{ij}^1, \rho_{ij}^2, \ldots, \rho_{ij}^k\right] \tag{10}$$

## 4. MSGWO-MKL Algorithm

### 4.1. Model of MKL

For one pair of time series, $\{\xi_i(v_i, t)\}_{t=1}^T$ and $\{\xi_j(x_{ij}, t)\}_{t=1}^T$, about UAV-*i* and UAV-*j*, the method of time series processing is able to transform them into a $1 \times k$ dimensional matrix $\rho_{ij}$.

Assuming that there exist such two clusters, $S_0$ and $S_1$, the two clusters were defined in Equation (11).

$$\begin{cases} S_1 = \left\{ \left[\rho_{ij}^1, \rho_{ij}^2, \ldots, \rho_{ij}^k, \bar{d}\right] \middle| a_{ij} = 1 \right\} \\ S_0 = \left\{ \left[\rho_{ij}^1, \rho_{ij}^2, \ldots, \rho_{ij}^k, \bar{d}\right] \middle| a_{ij} = 0 \right\} \end{cases} \tag{11}$$

where $\rho_{ij}^k$ is the k-order cross correlativity coefficient of one pair of time series, $\{\xi_i(v_i, t)\}_{t=1}^T$ and $\{\xi_j(x_{ij}, t)\}_{t=1}^T$, about UAV-*i* and UAV-*j*. The average distance of UAV-*i* and UAV-*j* in the time window is $\bar{d}$. The two clusters, $S_0$ and $S_1$, in an ideal kernel space are represented in Figure 4.

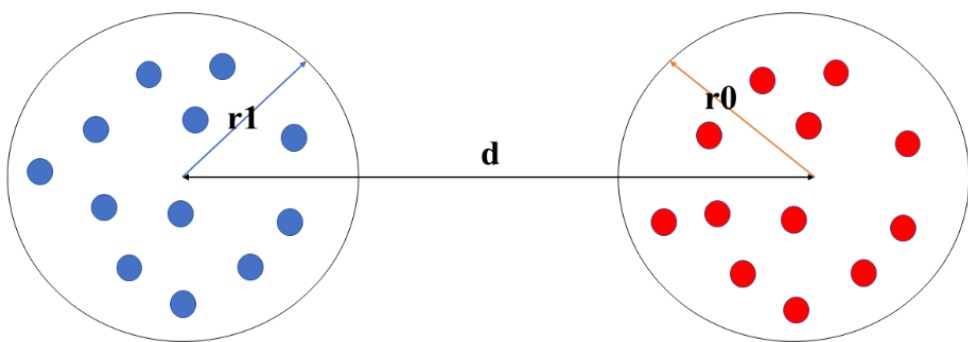

**Figure 4.** Clusters in an ideal kernel space.

The, $r_1$, is the character radius of the cluster $S_1$ in the ideal kernel space, $r_0$ is the character radius of the cluster, $S_0$, in the ideal kernel space, and $d$ is the character distance between the cluster, $S_0$, and the cluster, $S_1$, in the ideal kernel space.

Equation (12) was established in the ideal kernel space, predicting the missing links of UAV swarm networks.

$$a_{ij} = \begin{cases} 1, & if \ \| \phi(\rho_{ij}) - \phi\left(\rho_{center-(1)}\right) \| \leq r_1 \\ 0, & if \ \| \phi(\rho_{ij}) - \phi\left(\rho_{center-(0)}\right) \| \leq r_0 \end{cases} \tag{12}$$

where, $\rho_{center-(1)}$ and $\rho_{center-(0)}$, are the center points of the two clusters, $S_0$ and $S_1$, in the ideal kernel space. $\phi(\ )$ is the mapping function of the ideal kernel functions.

Traditional indicators of Kernel function are easy to fall into a local optimum with a slow convergence speed in the process of kernel learning, such as CSK [42,43] and KTA [44], passing through all the elements of the training set and operation's computational complexity to achieve $n^2$. To solve these problems, new indicators based on the idea of clusters were proposed.

The larger the character distance of clusters $d$, and the narrower the character radius, $r_1$ and $r_0$, are, the stronger the kernel function's classification capacity is. The computational complexity of evaluating kernel functions was reduced from $n^2$ to $(2n + 1)$, which improved computation efficiency significantly.

The character radius, $r_1$ and $r_0$, were calculated in Equation (13).

$$\begin{cases} r_1 = \frac{1}{n_1} \sum_{i=1}^{n_1} d_K\left(\vec{S_1}[i], \vec{S}_{1-center}\right) \\ r_0 = \frac{1}{n_0} \sum_{i=1}^{n_0} d_K\left(\vec{S_0}[i], \vec{S}_{0-center}\right) \end{cases} \tag{13}$$

where, $n_1$ and $n_0$, are the number of elements in the cluster $S_1$ and $S_0$, $\vec{S_1}[i]$ is the $i$-th element of the cluster $S_1$, $\vec{S_2}[i]$ is the $i$-th element of the cluster $S_2$, $\vec{S}_{1-center}$ is the center of the cluster $S_1$, $\vec{S}_{2-center}$ is the center of the cluster $S_2$, and $d_K$ is the distance in the kernel space.

The centers of the cluster, $S_1$ and $S_0$, were calculated in Equation (14).

$$\begin{cases} \vec{S}_{1-center} = \frac{1}{n1} \sum_{i=1}^{n1} \vec{S_1}[i] \\ \vec{S}_{0-center} = \frac{1}{n0} \sum_{i=1}^{n0} \vec{S_0}[i] \end{cases} \tag{14}$$

The distance in the kernel space, $d_K$, was calculated, in Equation (15).

$$d_K(X, Y) = Kernel(X, X) + Kernel(Y, Y) - 2\, Kernel(X, Y) \tag{15}$$

where $Kernel()$ is the kernel function, and $X$ and $Y$ are vectors in the Euclidean space.

The character distance $d$ of the cluster, $S_1$, and cluster, $S_0$, was calculated in Equation (16).

$$d = d_H\left(\vec{S}_{1-center}, \vec{S}_{0-center}\right) \tag{16}$$

Eventually, the process of multi-kernel learning was transformed into a multi-objective optimization problem in Equation (17).

$$\begin{cases} max\{f1[Kernel(\omega, \sigma)], f2[Kernel(\omega, \sigma)]\} \;.st. \; \| \omega_i \|_p \leq 1 \\ f1 = -(r_1 + r_0)/2, f2 = d \end{cases} \tag{17}$$

where $f1()$ and $f2()$ are the objective functions and $Kernel()$ is the kernel function to be optimized.

The kernel function $Kernel()$ was defined in Equation (18).

$$Kernel() = \sum_{i=1}^{m} \omega_i \times \overline{Kernel}_i(\sigma_i) \tag{18}$$

where $\overline{Kernel}_i()$ is the basis kernel function; $\omega_i$ is the weight of the basis kernel function $\overline{Kernel}_i()$; $\sigma_i$ is the parameter of the basis kernel function $\overline{Kernel}_i()$, and $m$ is the number of the basis kernel function $\overline{Kernel}_i()$.

The optimization solution of $Kernel()$ could be expressed as a one-dimensional matrix $X$, as in Equation (19).

$$X = [\omega_1, \omega_2, \ldots, \omega_m; \sigma_1, \sigma_2, \ldots, \sigma_m]_{1 \times 2m} \tag{19}$$

*4.2. MSGWO Algorithm*

4.2.1. Grey Wolf Optimizer Algorithm (GWO)

The GWO is a swarm intelligence optimization algorithm based on the social structures and predation behaviors of the wolves [45]. The results of GWO are obviously superior to Particle Swarm Optimization (PSO) and Differential Evolution (DE) in 29 standard test functions.

The grey wolf swarms have a strict social hierarchy. The three wolves with the best performance are defined as leader wolves $\alpha$, $\beta$, and $\delta$, and the other wolves are defined as follower wolves $\omega$. The follower wolves update their positions according to the condition

of the leader wolves. The original optimization process of GWO is shown in Algorithm 1. The updating process of the wolf population is as shown in Figure 5.

---

**Algorithm 1:** Grey Wolf Optimizer

**1:** for *iter* in range ($iter_{max}$):
**2:**   for *i* in range (*n*):
**3:**     $C_k = 2 \times random(0,1), k = 1,2,3$
**4:**     $D_\alpha = C_1 A_\alpha - A_i(iter), D_\beta = C_2 A_\beta - A_i(iter), D_\delta = C_3 A_\delta - A_i(iter)$
**5:**     $K_i = (2 - iter/iter_{max}) \times [2 \times random(0,1) - 1], k = 1,2,3$
**6:**     $A_1 = A_\alpha - K_1 D_\alpha, A_2 = A_\beta - K_2 D_\beta, A_3 = A_\delta - K_3 D_\delta$
**7:**     $A_i(iter + 1) = (A_1 + A_2 + A_3)/3$
**8:**   end for
**9:** end for

---

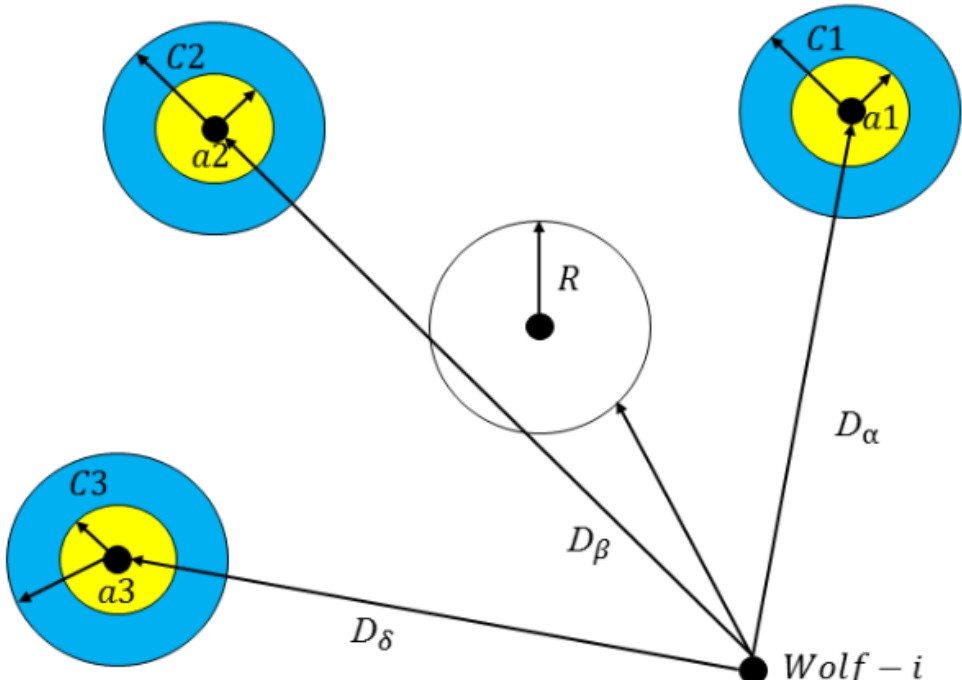

**Figure 5.** Updating process of wolves.

In Algorithm 1, $A_\alpha$, $A_\beta$, and $A_\delta$ represent the positions of the leader wolves $\alpha$, $\beta$, and $\delta$; $A_i(iter)$ represents the position of the wolf- *i* in the generation- *iter*; $D_\alpha$, $D_\beta$, and $D_\delta$ represents the search neighborhood generated by, $\alpha$, $\beta$, and $\delta$; $random(0,1)$ is a random number between (0,1), of which the randomness determines the uncertainty of the search neighborhood; *iter* represents the generation of the wolf packs; $iter_{max}$ represents the maximum generation; the parameter $K$ influences the wolf's neighborhood. If the absolute value of $K$ is more than one, the wolf swarms will face the neighborhood searching. If the absolute value of $K$ is less than one, the wolf swarms will leave the neighborhood to search. As shown in Figure 6, the uncertainty of the search neighborhood increases the global searching ability of the GWO.

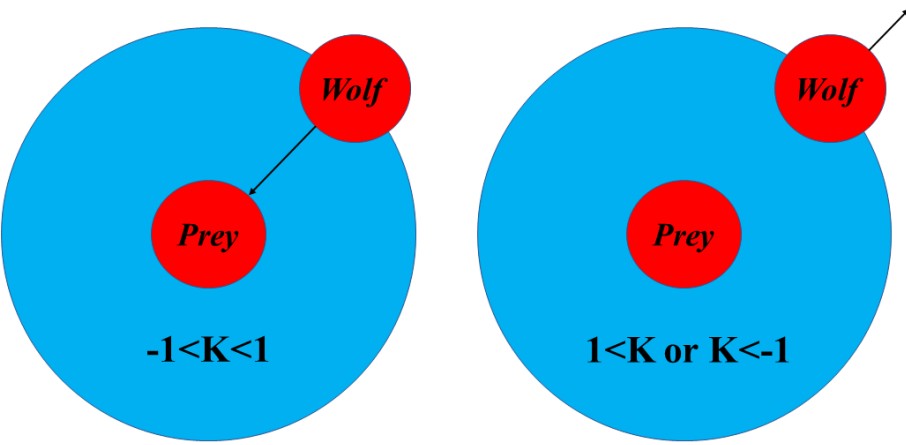

**Figure 6.** Search neighborhood.

### 4.2.2. Multi-Objective Design

This section modified the GWO algorithm into a multi-objective optimization algorithm. For multi-objective algorithms, the selection of leader wolves in each iteration is the most crucial step, which determines the superiority of the solutions directly.

Pareto Domination [46,47]: For multi-objective optimization algorithms, there are $n$ objective functions $f_i(x), i = 1, 2$. Given two decision variables, $X_a$ and $X_b$, if the two decision variables satisfy following Equation (20), then the decision variable $X_a$ is dominated by the decision variable $X_b$.

$$\begin{cases} f_i(X_a) \leq f_i(X_b), \forall\, i \in 1, 2. \\ f_i(X_a) < f_i(X_b), \exists\, i \in 1, 2. \end{cases} \tag{20}$$

where objective functions, $f_i(x), i = 1, 2$ aim to maximize.

If there exists a decision variable $X_{ND}$, which is not able to be dominated by other decision variables, then $X_{ND}$ is termed a non-dominated solution.

Pareto Rank: In a set of solutions, the non-dominated solutions $X_{ND}$ are termed rank one. Then the non-dominated solutions are removed from the original solution set. The non-dominated solutions, $X_{ND}$, in the new set of solutions are termed rank two. By analogy, the rank of all solutions in a solution set can be obtained, see Figure 7.

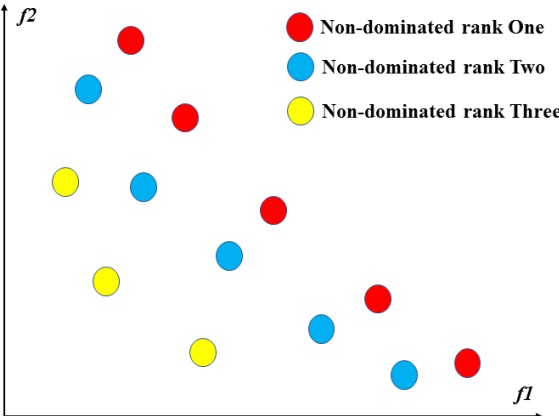

**Figure 7.** Non-dominated rank.

We selected leader wolves in the solution set of high non-dominated rank, according to the following Equation (21).

$$max\{RD = f_2(X_i)/(-f_1(X_i))\}, for\ X_i \epsilon\ Pareto \tag{21}$$

where *Pareto* is the solution set of high non-dominated rank, and *RD* is the relative distance between the character distance and character radius of two clusters.

The leader wolf selection process of MSGWO is shown in Algorithm 2.

---

**Algorithm 2:** Leader Wolves Selection

**1:** Define the scale of wolf population and position
**2:** for *i* in range(*n*):
**3:**     for *j* in range(*n*):
**4:**        if $f_i(X_a) \leq f_i(X_b), \forall i \in 1, 2$ and $f_i(X_a) < f_i(X_b), \exists i \in 1, 2.$:
**5:**          Then $X_a$ is dominated by $X_b$
**6:** Determine *Pareto* set
**7:** $max\{RD = f_2(X_i)/(-f_1(X_i))\}, for\ X_i \epsilon\ Pareto$
**8:** Determine the leader wolfs $\alpha, \beta$ and $\delta$
**9:** end

---

### 4.2.3. Adaptive Parameters Strategy

In the process of kernel learning, we found that the two objective functions, character radius, $f_1$, and character distance, $f_2$, were highly correlated and contradictory. Once the character distance reaches a high level, the character radius will also increase rapidly. Eventually, they all would be at a relatively high level whose performance was bad in standard data sets. Rigorous non-dominated rank hierarchy causes the local convergence of solutions in the process of kernel learning. The Weak Pareto Domination was proposed in Equation (22) to increase the diversity of leader wolves while preserving superior solutions in the next iteration.

Weak Pareto Domination: For multi-objective optimization algorithms, there are *n* objective functions, $f_i(x), i = 1, 2$. Given two decision variables, $X_a$ and $X_b$, if the two decision variables satisfy the following Equation (22), then the decision variable, $X_a$, is weak-dominated by the decision variable, $X_b$.

$$\begin{cases} f_1(X_a) < f_1(X_b)\ and\ (1 - \alpha) \times f_2(X_a) < f_2(X_b)\ (a) \\ (1 + \alpha)f_1(X_a) < f_1(X_b)\ and\ f_2(X_a) < f_2(X_b)\ (b) \end{cases} \tag{22}$$

where objective functions, $f_i(x), i = 1, 2$, aim to maximize; $\alpha$ is the adaptive parameter from 0 to 0.3, (*a*) and (*b*) correspond to the different two modules.

The Weak Pareto Domination was designed to encourage the wolf to explore while avoiding the local convergence at a relatively high level. The determination of adaptive parameters is in Equation (23)

$$\begin{cases} If\ |f_1(X_\alpha)| > k_1, then\ select\ (22.a)\ and\ let\ \alpha = 2 - |f_1(X_\alpha)| \\ If\ |f_2(X_\alpha)| < k_2, then\ select\ (22.b)\ and\ let\ \alpha = 2 \times |f_2(X_\alpha)| \end{cases} \tag{23}$$

where, *select* (22.*a*), means that selecting the (*a*) module of the Equation (22); $k_1$ and $k_2$, are experience factors, recommending, $k_1 = 1.7$, and $k_2 = 0.15$.

The Adaptive Parameters Strategy is shown in Algorithm 3.

---

**Algorithm 3:** Adaptive Parameters Strategy

**1:** Define the leader wolf $\alpha$
**2:** *if* $|f_1(X_\alpha)| > k_1$:
**3:**     Weak Pareto Domination-(a):
**4:**        $f_1(X_a) < f_1(X_b)\ and\ \ (1 - \alpha) \times f_2(X_a) < f_2(X_b)$
**5:**        $\alpha = 2 - |f1(X_\alpha)|$
**6:** *if* $|f_2(X_\alpha)| < k_2$:
**7:**     Weak Pareto Domination-(b):
**8:**        $(1 + \alpha) \times f_1(X_a) < f_1(X_b)\ and\ \ f_2(X_a) < f_2(X_b)$
**9:**        $\alpha = 2 \times |f_2(X_\alpha)|$
**10:** end

---

### 4.2.4. Opposition-Based Learning Strategy

The Opposition-Based Learning (OBL) strategy was proposed by H.Tizhoosh in 2005 [48,49], using opposite solutions, approximate opposite solutions, or inverse approximate opposite solutions to optimize the performance of an algorithm. It selects a total opposite solution based on the known best solution to increase the global searching ability. The method is widely used for algorithmic optimization in many algorithms.

In this section, the opposition-based learning strategy was adopted in the process of initializing the wolf population. We deleted three dominated wolves in the original wolf population and transformed the leader wolves into the opposite leader wolves, putting the opposite leader wolves into the new wolf population, which would increase the quality of leader wolves significantly and the diversity of the original population. The Opposition-based Learning (OBL) strategy is as Equation (24).

$$X_{OBL} = LB + UB - X_\alpha + r \cdot (X_\alpha - X) \tag{24}$$

where, $X_{OBL}$, is the solution generated by the Opposition-based Learning (OBL) strategy, $LB$ is the lowest limit solution, $LB$ is the uppermost limit solution, $X_\alpha$ is the solution of leader wolf $\alpha$, $X$ is the original solution, and $r$ is a random number from 0 to 1.

The Opposition-Based Learning strategy is shown in Algorithm 4.

---

**Algorithm 4**: Opposition-Based Learning Strategy
**1:** Define the wolf $i$
**2:** For $i$ in range ($n$):
**3:**    For $j$ in range (20):
**4:**        $X_{OBL}(i) = LB + UB - X_\alpha + r \cdot (X_\alpha - X(i))$
**5:**        IF ($X(i)$ is dominated by $X_{OBL}(i)$):
**6:**            $X(i) = X_{OBL}(i)$
**7:**            Break
**8:** end

---

### 4.2.5. Neighbor Learning Strategy

In addition to group hunting, individual hunting is another interesting social behavior of grey wolves, which is our motivation to improve the GWO [50]. The Neighbor Learning Strategy (NLS) was proposed in this section. In NLS, each individual wolf was motivated by its neighbors to find another strategy for a better position. The following steps describe how NLS search strategies generate better positions.

Radius $R_i(t)$ was calculated using Euclidean distance between the current position of $X_i(t)$ and the position $X_{i-GWO}(t+1)$ generated by GWO, in Equation (25).

$$R_i(t) = \| X_i(t) - X_{i-GWO}(t+1) \| (1.5)^n \tag{25}$$

where , $n = 1, 2, \ldots$, to adjust the neighborhood scope of $X_i(t)$.

Then, the neighborhood scope of $X_i(t)$ was constructed by Equation (26).

$$N_i(t) = \left\{ X_j(t) \middle| D_i(X_i(t), X_j(t)) < R_i(t) \right\} \tag{26}$$

where $N_i(t)$ is the neighborhood scope of the wolf $X_i(t)$; $D_i(X_i(t), X_j(t))$ is the Euclidean distance between $X_i(t)$ and $X_j(t)$.

$M_i(t)$ is the absolute complement of set $N_i(t)$, in Equation (27).

$$M_i(t) = \left\{ X_j(t) \middle| X_j(t) \notin N_i(t), X_j(t) \in \Omega \right\} \tag{27}$$

where $\Omega$ is a set containing all the wolves' positions.

Once the neighborhood scope, $N_i(t)$ and $M_i(t)$, are constructed, Neighbor Learning Strategy is performed by Equation (28).

$$X_{NLS}(i) = X_i + random \cdot (X_N(i) - X_M(i)) \tag{28}$$

where, $X_N(i)$, is a random element selected from $N_i(t)$, $X_M(i)$ is a random element selected from $M_i(t)$, *random* is a random number from 0 to 1, and $X_{NLS}$ is a solution generated by the Neighbor Learning Strategy.

The Neighbor Learning Strategy is shown in Algorithm 5.

---

**Algorithm 5**: Neighbor Learning Strategy

**1:** Define the wolf population
**2:** For $i$ in range ($n$):
**3:**　For $j$ in range (20):
**4:**　　For $n$ in range (100):
**5:**　　$R_i(t) = \| X_i(t) - X_{i-GWO}(t+1) \| (1.5)^n$
**6:**　　　$N_i(t) = \left\{ X_j(t) \middle| D_i\left(X_i(t), X_j(t)\right) < R_i(t) \right\}$
**7:**　　　IF $N_i(t) \neq \varnothing$:
**8:**　　　　Break
**9:**　　　$M_i(t) = \left\{ X_j(t) \middle| X_j(t) \notin N_i(t), X_j(t) \in \Omega \right\}$
**10:**　　　$X_{NLS}(i) = X_i + random \cdot (X_N(i) - X_M(i))$
**11:**　　IF ($X(i)$ is dominated by $X_{NLS}(i)$):
12 :　　　$X(i) = X_{NLS}(i)$
**13:**　　　Break
**14:** end

---

### 4.2.6. Shuffle Learning Strategy

The Shuffle Learning strategy creates a new neighborhood by the shuffle method. The illumination of the Shuffle Learning strategy is shown in Figure 8.

**Figure 8.** Shuffle Learning Strategy.

When the Opposition-Based Learning strategy and Neighbor Learning strategy are not able to generate superior solutions, it is time that shuffle learning is adopted. The shuffling operation will generate a random sequence, forming a set of new positions. Although the shuffling operation will explore considerable feasible solutions and improve the global searching ability, it will also destroy the original structure of the neighborhood and miss the superiority of leader wolves.

The Shuffle Learning strategy is shown in Algorithm 6.

---

**Algorithm 6**: Shuffle Learning Strategy

**1:** Define the wolf population
**2:** For $i$ in range ($n$):
**3:**　For $j$ in range (20):
**4:**　　$X_{SLS}(i) = np.random.shuffle(X[i])$
**5:**　　IF ($X(i)$ is dominated by $X_{SLS}(i)$):
6 :　　　$X(i) = X_{NLS}(i)$
**7:**　　　Break
**8:** end

---

### 4.2.7. Flow of the MSGWO

The flow of the MSGWO algorithm is shown in Figure 9. The red part means the components of the original GWO, the blue part means the judgement module, and the yellow component are improved in this paper.

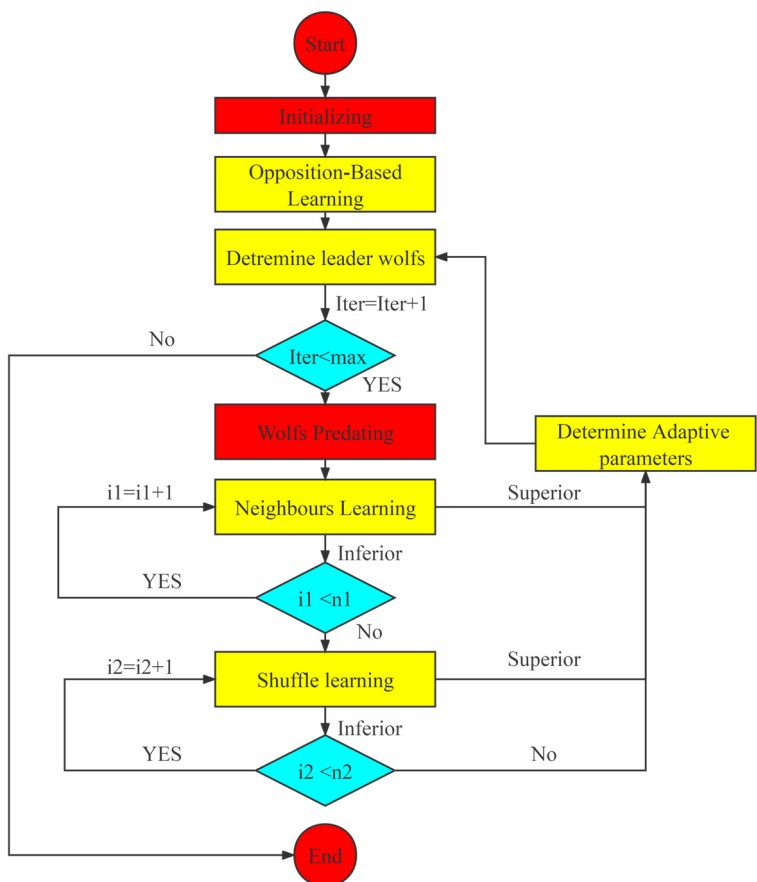

**Figure 9.** Flow of MSGWO.

When the MSGWO algorithm is launched, the initializer will generate the initial population of the grey wolves randomly. The weak Pareto rank norm will select leader wolves. Then we delete three dominated wolves from the original wolf population and transform the leader wolves into the opposite leader wolves, putting the opposite leader wolves into the new wolf population, which will increase the quality of leader wolves significantly and the diversity of the original population.

After that, the original GWO algorithm will generate a set of solutions to the multi-objective optimization problem. However, the GWO algorithms are easily trapped in local optimization, and the quality of solutions generated by the GWO algorithm should be enhanced further. To solve those problems, the neighbors' learning strategy and shuffle learning strategy are adopted. New neighborhoods generated by the neighbors' learning strategy and shuffle are able to increase the global searching ability of the GWO evidently. Once the solution generated by the neighbors learning strategy dominates the original solution, the shuffle learning strategy will be skipped directly. Eventually, the adaptive parameters will be adjusted according to the conditions of the leader wolves.

## 5. MSGWO-MKL-SVM Algorithm

As a general method for classification proposed by Vapnik [51,52], the support vector machine essentially uses a kernel function that maps the original input data space into a high-dimensional feature space so that the instances from two classes are as far apart as possible, preferably separable with a linear boundary in a Hilbert space.

Given a sample, $\{x_i, y_i; i = 1, \ldots, n\}$, where $x_i$ is a vector of predictors in the input space and $y_i$ represents the class index, which takes a value from $\{+1, -1\}$, a nonlinear support vector machine maps the input data $\{x_1, x_2, \ldots, x_n\}$ into a high-dimensional feature space, using a nonlinear mapping function $\phi$, and finds a linear boundary in the feature space by maximizing the smallest distance of instances to this boundary. Mathematically, the idea is equivalent to solving the Equation (29).

$$\begin{cases} max \sum_{i=1}^n \alpha_i - \frac{1}{2} \sum_{i=1}^n \sum_{j=1}^n \alpha_i \alpha_j y_i y_j K(x_i, x_j) \\ subject\ to\ \sum_{i=1}^n \alpha_i y_i = 0, 0 \le \alpha_i \le C, i = 1, 2, \ldots n \end{cases} \tag{29}$$

where $\alpha_i$ is the dual variable and the scalar function $K(x_i, x_j)$ and is called a kernel function, adopting the multiple kernel functions learned in the last section.

The kernel form of the SVM boundary can be written as Equation (30).

$$\sum_{i \in SV} \alpha_i y_i K(x_i, x) + b = 0 \tag{30}$$

where $SV$ is the set of support vectors.

We put the kernel functions optimized by the MSGWO algorithm into the SVM algorithm, forming the complete MSGWO-MKL-SVM algorithm.

## 6. Numerical Experiment

To illustrate the effectiveness of the MSGWO-MKL-SVM algorithm proposed in this paper, we designed three experiments:

1. Testing the multi-objective optimization ability of the MSGWO algorithm;
2. Testing the classification ability of the MSGWO-MKL algorithm;
3. Testing the missing links predicting ability of the MSGWO-MKL-SVM algorithm.

All experiments were conducted with Python 3.6, running on an Intel Core i7-8565CPU @ 1.80 GHz, and Windows 7 Ultimate Edition.

### 6.1. Multi-Objective Optimization Ability Test of MSGWO

This section selected the Blood Transfusion dataset in UCI to examine the MO-IGWO's optimization in the process of multi-objective optimization. The objective functions were adopted as in Equation (17). The relative distance ($RD$) is the ratio of character distance ($CD$) and character radius ($CR$) of clusters in kernel space. The MO-PSO [53], MO-GWO [54], and NSGA-2 [46] algorithms were used for comparison, demonstrating the superiority of the MSGWO algorithm.

As shown in Table 1, in all experiments, the parameters of the comparative algorithms were the same as the recommended settings.

**Table 1.** Parameters of algorithms.

|  | **MO-PSO** | **MO-GWO** | **NSGA-2** | **MSGWO** |
|---|---|---|---|---|
| Population | 20 | 20 | 100 | 20 |
| Generation | 150 | 150 | 200 | 50 |

The optimization process of MO-PSO is as shown in Figure 10, and the relative distance of MO-PSO is as shown in Figure 11. The optimization process of NSGA-2 is as shown in Figure 12 and the relative distance of NSGA-2 is as shown in Figure 13. The optimization process of MO-GWO is as shown in Figure 14 and the relative distance of MO-GWO is as shown in Figure 15. The optimization process of MSGWO is as shown in Figure 16 and the relative distance of MSGWO is as shown in Figure 17.

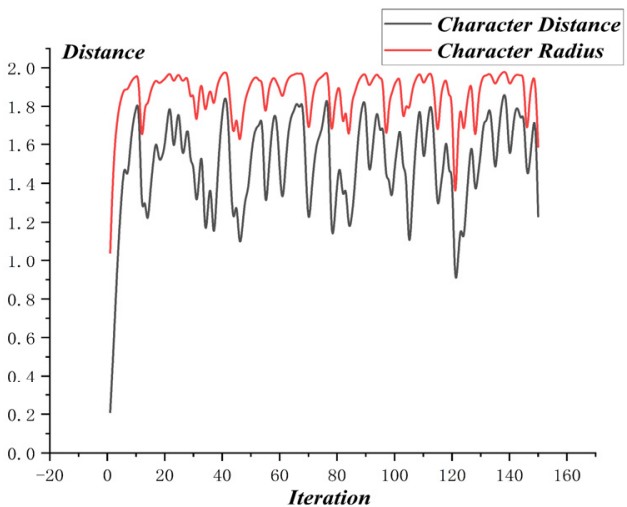

**Figure 10.** Optimization process of MO-PSO.

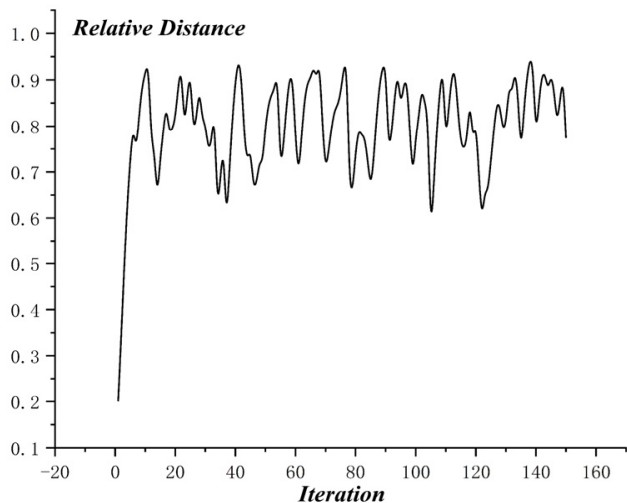

**Figure 11.** Relative distance of MO-PSO.

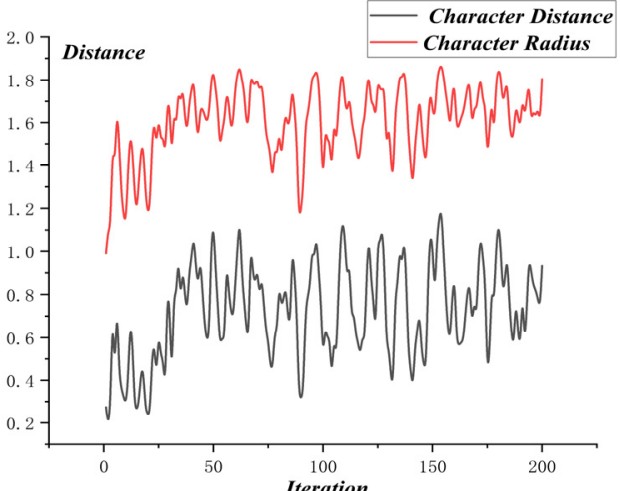

**Figure 12.** Optimization process of NSGA-2.

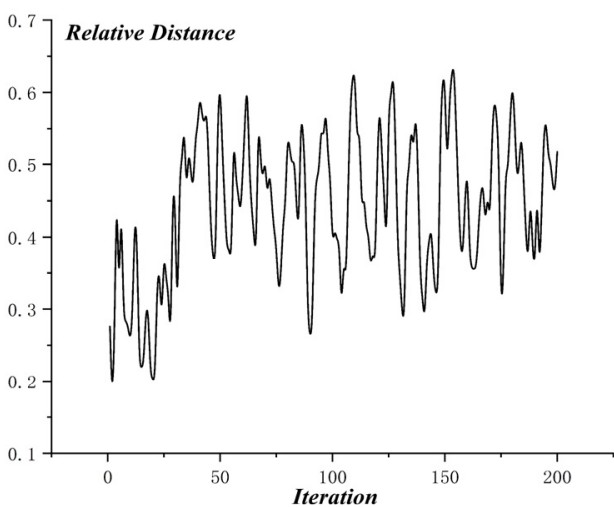

**Figure 13.** Relative distance of NSGA-2.

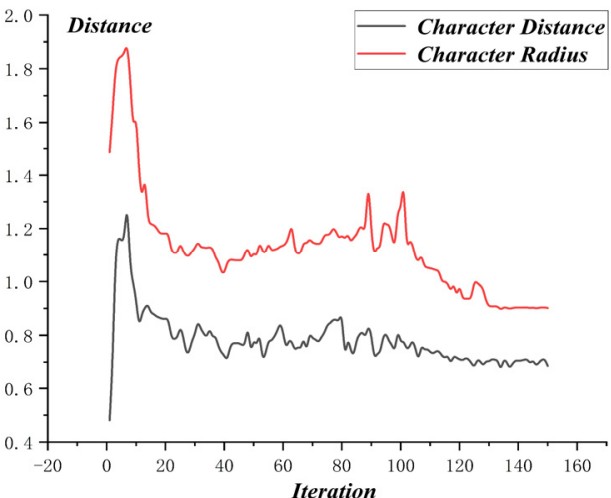

**Figure 14.** Optimization process of MO-GWO.

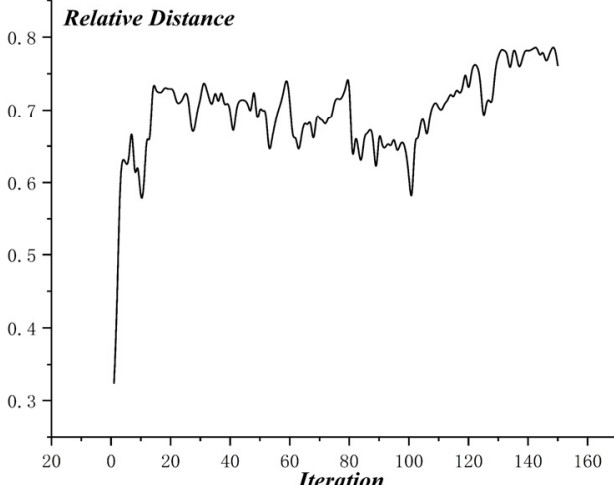

**Figure 15.** Relative distance of MO-GWO.

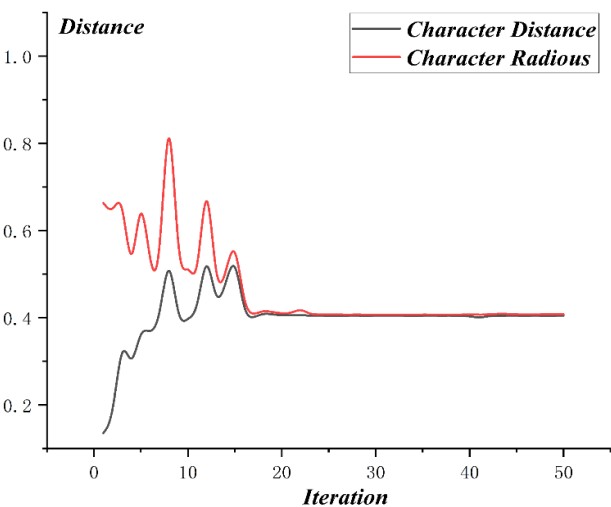

**Figure 16.** Optimization process of MSGWO.

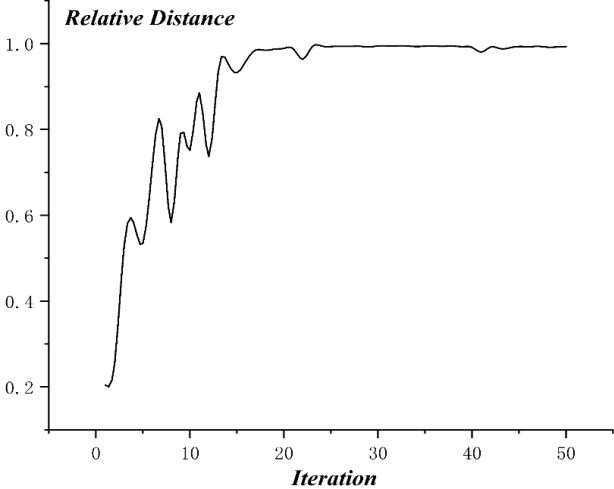

**Figure 17.** Relative distance of MSGWO.

In the optimization process of MO-PSO, the character distance and character radius both showed an increasing tendency, with considerable oscillation. In the finite 150 iterations, the algorithm MO-PSO didn't reach the state of convergence. However, the relative distance of MO-PSO was kept at a high level constantly, with slight oscillation.

In the optimization process of NSGA-2, the character distance showed an increasing tendency, and the character radius was kept at a low level. In the finite 200 iterations, the algorithm NSGA-2 didn't reach the state of convergence. The relative distance of NSGA-2 was kept at around 0.5, with violent oscillation.

In the optimization process of MO-GWO, the character distance and character radius vary almost simultaneously, and they both show a decreasing tendency. In the finite 150 iterations, the algorithm MO-GWO didn't reach a state of convergence. However, the relative distance of MO-GWO was kept at a high level constantly, with a tendency of convergence.

In the optimization process of MSGWO, the character distance shows an increasing tendency, and the character radius shows a decreasing tendency. In the finite 20 iterations, the algorithm MSGWO reached a state of convergence. The relative distance of MSGWO was kept at a high level constantly, without oscillation.

Then, the kernel functions optimized by the above algorithms were submitted into the SVM method, solving a classification problem in the Blood Transfusion data set. The results

of Character Distance, Character Radius, Relative Distance and classification accuracy are shown in Table 2.

**Table 2.** Classification Results.

|  | MO-PSO | NSGA-2 | MO-GWO | MSGWO |
|---|---|---|---|---|
| Character Distance | 1.63 | 0.93 | 0.686 | 0.404 |
| Character Radius | **1.870** | 1.810 | 0.901 | 0.407 |
| Relative Distance | 0.869 | 0.513 | 0.761 | 0.993 |
| Accuracy | 62.4% | 68.0% | 67.1% | 78.2% |

For *Character Distance*, the MSGWO algorithm reached the minimum value of 0.404. For *Character Radius*, the MO-PSO algorithm reached the maximum value of 1.870. For *Relative Distance*, the MSGWO algorithm reached the maximum value of 0.993. It is not difficult to find that the MO-PSO and NSGA-2 algorithms are easy to increase *Character Distance* and *Character Radius* simultaneously, and the MO-GWO algorithm is easy to decrease *Character Distance* and *Character Radius* simultaneously. When the two objective functions, like *Character Distance* and *Character Radius* are conflicting and highly coupled, the MSGWO could balance the contradiction of different objective functions in the process of multi-objective optimization. From the number of iterations, the MSGWO algorithm used fewer iterations and obtained superior solutions, which is sufficient to demonstrate that the MSGWO algorithm has stronger global searching ability and faster convergence speed compared with the other state-of-the-art algorithms in this given multi-objective optimization problem. The classification accuracy in the Blood Transfusion data set also demonstrated the effectiveness of the MSGWO algorithm.

### 6.2. Classification Ability Test of MSGWO-MKL

The kernel function optimized by MSGWO was put into the SVM algorithm to verify the classification ability of the MSGWO-MKL algorithm. We repeated ten fold cross-validation 20 times in eight standard UCI data sets, comparing with other state-of-the-art SVM classification algorithms, such as SVM, TML-SVM [55], ISSML-SVM [56], LMN-SVM [57], and PCML-SVM [58]. The classification accuracy of eight UCI data sets is shown in Table 3.

**Table 3.** Classification Results.

| Standard Dataset | SVM | TML-SVM | ISSML-SVM | LMNN-SVM | PCML-SVM | MSGWO-MKL |
|---|---|---|---|---|---|---|
| Blood Transfusion | 65.3% | 69.2% | 68.3% | 71.3% | 70.9% | 78.2% |
| Breast Cancer | 75.3% | 75.7% | 76.5% | 73.1% | 74.9% | 77.8% |
| German | 74.8% | 75.5% | 75.7% | 75.2% | 76.4% | 84.1% |
| Heart | 85.8% | 86.5% | 86.0% | 85.4% | 85.1% | 85.7% |
| Liver | 74.0% | 75.6% | 74.5% | 72.5% | 73.3% | 79.0% |
| Parkinson | 85.8% | 88.2% | 89.5% | 88.2% | 88.5% | 96.1% |
| Pima | 77.0% | 77.5% | 78.7% | 77.5% | 78.2% | 79.8% |
| Sonar | 86.0% | 86.5% | 84.6% | 83.7% | 85.5% | 92.6% |

According to the results of Table 3, the classification accuracy of MSGWO-MKL was improved by 6.2% on average compared with the original SVM algorithm in eight standard UCI data sets. For the data set Blood Transfusion, German and Parkinson, the classification accuracy of MSGWO-MKL was increased by more than 10% compared with the best result. For the data set Breast Cancer, Liver, Pima, and Sonar, the classification accuracy of MSGWO-MKL was increased by less than 5% compared with the best result. For the data set Heart, the difference between the classification accuracy of all algorithms is within 1%.

The test results in eight standard UCI datasets proved that the kernel function optimized by the MSGWO algorithm could help the SVM algorithm increase the classification

accuracy to some extent. The results also showed that the classification accuracy increments caused by the MSGWO-MKL algorithm were related to the given data set. The performance of the MSGWO-MKL algorithm differed widely in different data sets.

### 6.3. Missing Link Predicting Ability Test of MSGWO-MKL-SVM

When a UAV swarm was carrying out missions to check out or carry out a strike on adversarial airports, they would be confronted with a variety of highly intensive anti-air weapons and would be easily intercepted without any countermeasures. The formation control of a UAV swarm in enemy airspace is a common tactical movement. The time series data is observed from the process of formation control of a UAV swarm. Figure 18 depicts the completion of the simulation process in the Any-logic simulation platform. The topology structure of the UAV network was generated randomly, as in Equation (31).

$$A_{UAV} = \left[ a_{ij}^{UAV} \right] = \begin{cases} 0, \; if \; x > P_0 \\ 1, \; if \; x \leq P_0 \end{cases} \tag{31}$$

where $A_{UAV}$ is the adjacency matrix of the UAV swarm, $a_{ij}^{UAV}$ is the element of the matrix, $A_{UAV}$, $x$ is a random number from 0 to 1, and $P_0$ is the probability parameter of two UAVs having communication links, recommending that $P_0 = 0.5$.

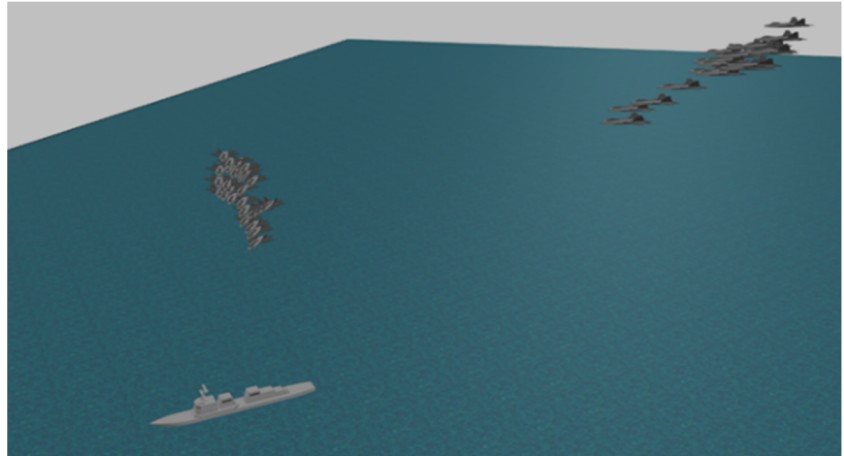

**Figure 18.** The simulation in Any-logic platform.

We repeated the simulation of UAV swarm formation control 500 times on the Any-logic platform, obtaining the time series of the UAV swarm network in a given topology network. Seven different scenarios were designed to verify the missing link predicting ability of the MSGWO-MKL-SVM algorithm. The conditions of different scenarios are shown in Table 4.

**Table 4.** Conditions of different scenarios.

| Scenario | UAV Number | Observation Time Length/s |
|:---:|:---:|:---:|
| 1 | 20 | 10 |
| 2 | 30 | 10 |
| 3 | 50 | 5 |
| 4 | 50 | 10 |
| 5 | 50 | 15 |
| 6 | 50 | 20 |
| 7 | 70 | 10 |

Then the ten fold cross-validation was repeated 120 times. The other state-of-the-art MLP algorithms, such as Common Neighbors (CN) [3] and Amplitude Difference Method (ADM) [18], were compared with the proposed MSGWO-MKL-SVM algorithm.

Link prediction results are shown in Figure 19A–G and Table 5. Figure 19A–G depicts a series of scatter diagrams corresponding to Scenarios 1–7.

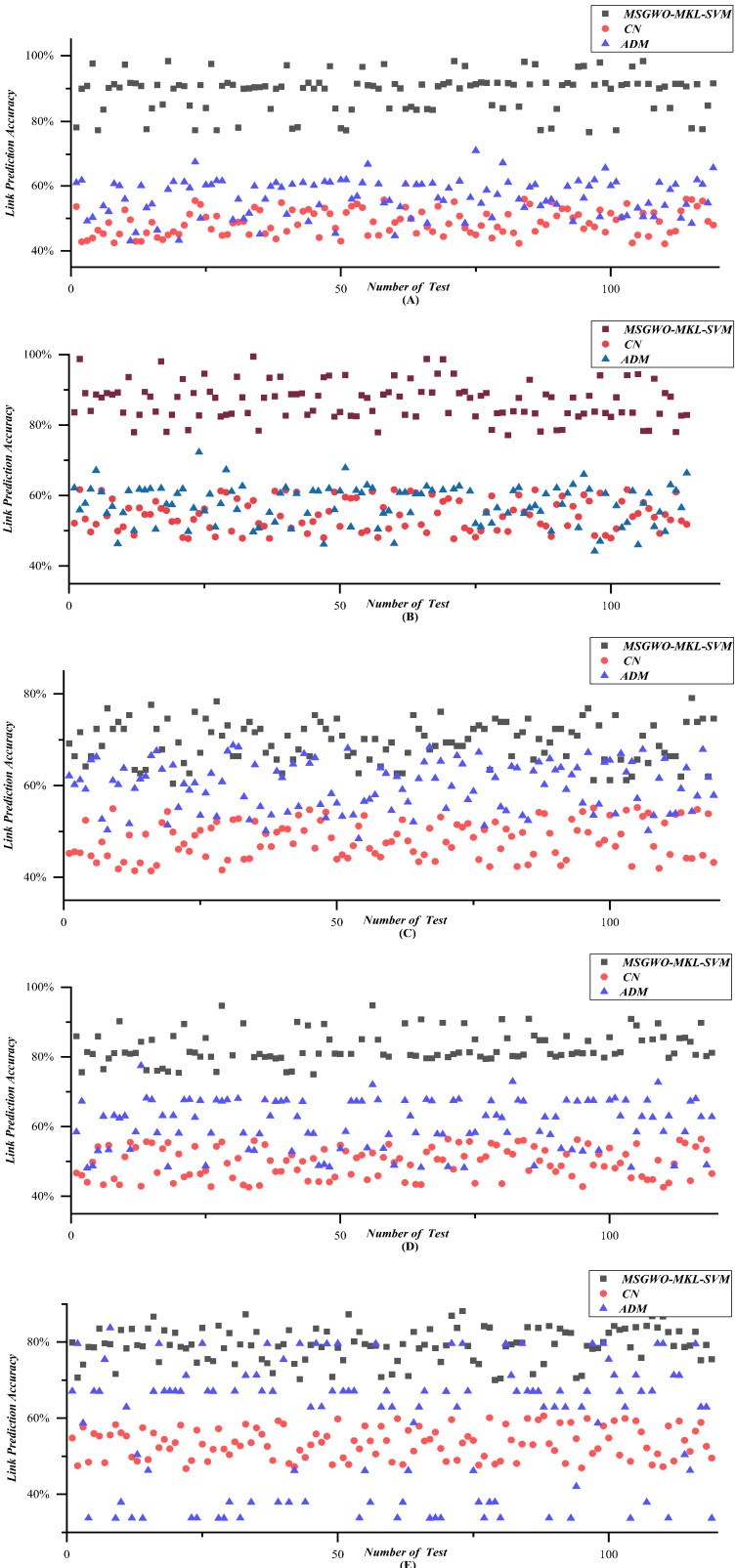

**Figure 19.** *Cont.*

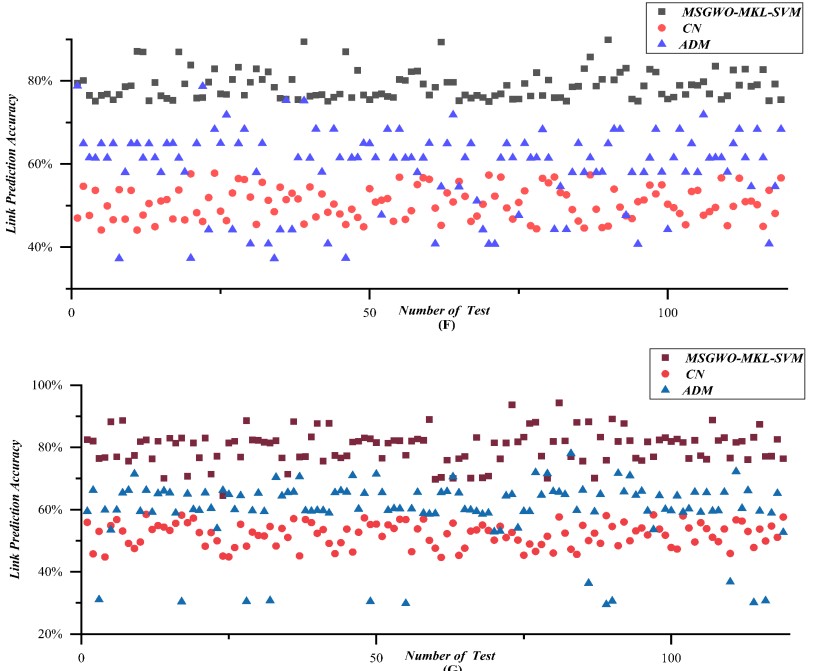

**Figure 19.** (**A**–**G**) Scatter diagrams of link prediction results.

**Table 5.** Results of link prediction.

| Scenario | Accuracy | CN | ADM | MSGWO-MKL |
|----------|----------|-----|-----|-----------|
| Scen1 | Average | 48.62% | 56.50% | 89.03% |
| | Std.dev | $3.84 \times 10^{-2}$ | $5.76 \times 10^{-2}$ | $5.77 \times 10^{-2}$ |
| | Median | 48.53% | 57.31% | 90.07% |
| | Maximum | 55.95% | 71.11% | 98.33% |
| | Minimum | 42.15% | 43.03% | 76.70% |
| Scen2 | Average | 54.52% | 57.75% | 86.78% |
| | Std.dev | $4.42 \times 10^{-2}$ | $5.48 \times 10^{-2}$ | $5.33 \times 10^{-2}$ |
| | Median | 54.37% | 57.76% | 87.75% |
| | Maximum | 61.68% | 72.27% | 99.42% |
| | Minimum | 47.73% | 44.22% | 77.09% |
| Scen3 | Average | 48.26% | 59.99% | 69.40% |
| | Std.dev | $3.98 \times 10^{-2}$ | $5.58 \times 10^{-2}$ | $4.30 \times 10^{-2}$ |
| | Median | 48.02% | 59.72% | 69.40% |
| | Maximum | 55.24% | 74.27% | 78.35% |
| | Minimum | 41.40% | 46.22% | 60.44% |
| Scen4 | Average | 49.68% | 60.82% | 82.67% |
| | Std.dev | $4.33 \times 10^{-2}$ | $7.09 \times 10^{-2}$ | $4.44 \times 10^{-2}$ |
| | Median | 50.24% | 62.72% | 81.06% |
| | Maximum | 56.42% | 77.45% | 94.73% |
| | Minimum | 42.68% | 48.13% | 74.99% |

**Table 5.** *Cont.*

| Scenario | Accuracy | CN | ADM | MSGWO-MKL |
|----------|----------|-----|-----|-----------|
| Scen5 | Average | 53.51% | 58.58% | 79.25% |
| | Std.dev | $4.06 \times 10^{-2}$ | $1.66 \times 10^{-1}$ | $4.47 \times 10^{-2}$ |
| | Median | 53.63% | 67.01% | 79.24% |
| | Maximum | 60.59% | 83.71% | 88.11% |
| | Minimum | 46.75% | 33.68% | 70.01% |
| Scen6 | Average | 50.05% | 59.20% | 78.85% |
| | Std.dev | $3.86 \times 10^{-2}$ | $9.55 \times 10^{-2}$ | $3.49 \times 10^{-2}$ |
| | Median | 50.63% | 61.47% | 78.46% |
| | Maximum | 57.79% | 78.75% | 89.86% |
| | Minimum | 44.11% | 37.28% | 75.02% |
| Scen7 | Average | 51.94% | 59.99% | 80.09% |
| | Std.dev | $3.82 \times 10^{-2}$ | $1.04 \times 10^{-1}$ | $5.45 \times 10^{-2}$ |
| | Median | 52.66% | 60.31% | 81.64% |
| | Maximum | 58.47% | 78.04% | 84.29% |
| | Minimum | 44.64% | 60.31% | 64.46% |

According to the results of Figure 19A–G, and Table 5, the link prediction accuracy was improved by 25.9% by MSGWO-MKL-SVM, compared with the average results, which strongly demonstrated the validity of the MSGWO-MKL-SVM algorithm. The results also showed that when facing a UAV swarm network with strong randomness and high uncertainty, the similarity-based method CN performed badly. Although the ADM algorithm that coped with one-dimensional time series performed better than the similarity-based method CN, it was still at a low level. Besides that, the link prediction accuracy of the ADM algorithm is very unstable, especially for the scenarios of a large number of UAVs and long-time observation. The MSGWO-MKL-SVM algorithm has obvious advantages in both link prediction accuracy and stability.

The variation of prediction accuracy about UAV number is shown in Figure 20, and the variation of prediction accuracy about observation time length is shown in Figure 21.

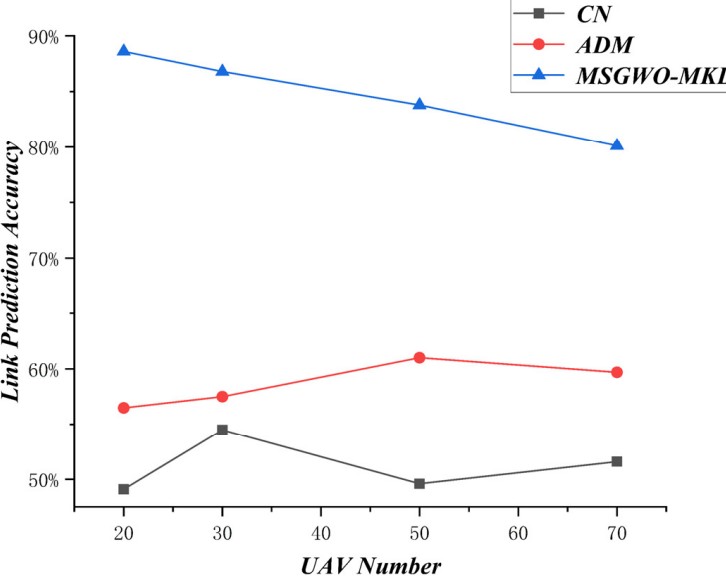

**Figure 20.** Variation of accuracy about UAV number.

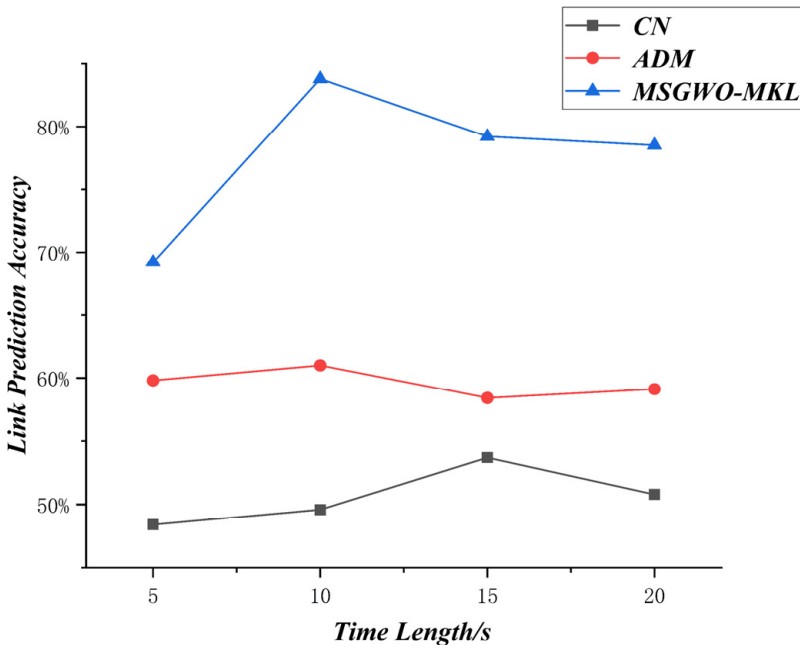

**Figure 21.** Variation of accuracy about time length.

According to the results of Figure 20, with the scale of the UAV swarm increasing, the missing link prediction accuracy of the MSGWO-MKL-SVM algorithm showed a downward trend. This is due to the fact that, with the scale of the UAV swarm increasing, the UAV swarm network will get more and more complicated, and the difficulty of link prediction will also increase at the same time.

According to the results of Figure 21, with the observation time length increasing, the missing link prediction accuracy of the MSGWO-MKL-SVM algorithm showed an upward trend and then a downward trend. This is due to the fact that, with the observation time length increasing, the information in the time series is increasing, and the difficulty of link prediction will also decrease at the same time. However, a UAV swarm is a dynamic, time-varying, complex network. If the observation time is too long, the UAV swarm network will undergo many changes, resulting in part of the time series data being invalid.

## 7. Discussion

This paper proposed a new algorithm, MSGWO-MKL-SVM, based on time series to solve the MLP problems of UAV swarm networks. In Section 6, the multi-objective optimization ability of the MSGWO algorithm, the classification ability of the MSGWO-MKL algorithm, and the missing links predicting ability of the MSGWO-MKL-SVM algorithm have been verified properly by a series of comparative experiments. Although the MSGWO-MKL-SVM algorithm exhibited excellent performance, the following several flaws remain when dealing with MLP problems of UAV swarm networks.

1.  In the process of multi-kernel learning, the basement set of basic kernel functions is lack of completeness proof. However, in the absence of proper determination methods, the selection of basic kernel functions is heavily reliant on the experience of researchers. This means that the performance of the MSGWO-MKL algorithm may fluctuate in different scenarios. The investigation into the completeness of basic kernel functions should be brought to the forefront of the field of multi-kernel learning;

2.  In the process of multi-kernel learning, heuristic algorithms were adopted to optimize the kernel functions. The calculating cost increases exponentially with the amount of data. The computational complexity of evaluating a kernel function's performance is $o(n^2)$, where, $n$, is the amount of data. The computational complexity of multi-objective optimization is $o(s!)$, where, $s$, is the scale of solution space, so the computational complexity of the MSGWO-MKL algorithm is $n^2 \times o(s!)$. The new indicators

of MKL transform the computational complexity of evaluating a kernel function's performance from $o(n^2)$ to $o(2n+1)$. The assumption is that the MSGWO algorithm transforms the computational complexity of the multi-objective optimization problem from $o(s!)$ to $o\left(s^k\right)$, $k$ is a natural number, and that the computational complexity of the MSGWO-MKL algorithm is also $(2n+1) \times o\left(s^k\right)$. High computational complexity is a common problem in the process of multi-kernel learning, and there is no doubt that certain effective optimization algorithms with low computational complexity are urgently needed, such as some convex optimization algorithms;

3. In the real battlefield environment, the enemy UAV swarm do not tend to conduct formation control frequently and actively. This means that the effective information of the observing time series data is finite. The defenders should apply more electronic interference to the enemy UAV swarm, compelling enemy UAV swarms to enhance the frequency of formation control so that the observing time series data could contain more information that is effective.

## 8. Conclusions

Many developed countries are investigating and developing AUDT technologies to defend against advanced enemy UAV swarm attacks in the future battlefield environment. Accurate link prediction of UAV swarm networks can help the defender quickly generate an efficient dynamic weapon target assignment scheme to disintegrate the enemy UAV swarm at a very low cost.

This paper proposed a new algorithm, MSGWO-MKL-SVM, based on time series to solve the MLP problems of UAV swarm networks. In this paper, $k$-order cross-correlation features of time series data about UAV swarm networks were extracted. Then we introduced the multi-kernel learning (MKL) techniques into the process of link prediction of the UAV swarm network, and the multi-strategy grey wolf optimizer algorithm (MSGWO) was adopted to optimize the kernel function. The MSGWO algorithm can effectively avoid local convergence in the process of multi-objective optimization, while the proposed indicator of MKL based on cluster greatly reduces the computational complexity. In the end, the multi-objective optimization ability of the MSGWO algorithm, the classification ability of the MSGWO-MKL algorithm, and the missing links predicting ability of the MSGWO-MKL-SVM algorithm have been verified properly by a series of comparative experiments. Meanwhile, the proposed method can be used for some link prediction on some social networks as well. From the experimental results and discussions, the following conclusions can be drawn:

1. The $k$-order cross-correlativity coefficient used in this paper could quantify the influence degree between nodes effectively. Then, the MSGWO-MKL-SVM algorithm could calculate the threshold hyperplane of direct influence and indirect influence based on time series, distinguishing 1-hop nodes and $k$-hop nodes validly;
2. The observation length of time series should be moderate. Too long or too short, all will impair the link prediction accuracy of UAV swarm networks;
3. New indicators of multi-kernel learning (MKL) based on clusters, transformed a multi-kernel learning problem into a multi-objective optimization problem and greatly reduced the computational complexity in the process of optimization;
4. The design of multi-strategy can enhance the balance between local and global search of the GWO algorithm and maintain diversity. In the standard UCI dataset, the MSGWO algorithm performed better than some state-of-the-art algorithms.

**Author Contributions:** Writing—original draft, M.N.; Writing—review & editing, Y.Z., J.Z., T.W. and X.Z. All authors have read and agreed to the published version of the manuscript.

**Funding:** National Natural Science Foundation of China: No. 72101263.

**Institutional Review Board Statement:** Not applicable.

**Informed Consent Statement:** Not applicable.

**Data Availability Statement:** Not applicable.

**Conflicts of Interest:** The authors declare no conflict of interest.

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
