# Peer review of "MSGWO-MKL-SVM: A Missing Link Prediction Method for UAV Swarm Network Based on Time Series"

_mathematics, doi:10.3390/math10142535_

Round 1

Reviewer 1 Report

The paper is well written and the proposed method is well described and sounding, even if it looks very complex. Too complex if we consider that improvement of performance wrt other algorithms already available is not that big.

Moreover, the comparisons are not presented in a statistically rigorous manner: presenting just the average over 20 runs does not give the right/appropriate picture of the performance. On top of this, the authors just present the results without really trying to explain why by correlating the features of the problems with those of the algorithms.

Conclusions should be rewritten once the data have been analyzed better.

Author Response

Dear reviewer:

Thank you in spite of being very busy to review my manuscript.Your comments are very helpful and thoughtful for me.According to your comments,there are following adjustments in my manuscript.Please see the attachment

Best wishes to you!

                                                                                                           2022.6.27

Reviewer 2 Report

The objective of the paper is not clear.

The purpose of “missing link prediction” in a UAV/drone swarm is not well-defined in the paper.

It is also not a well-known/well-defined problem in the scientific/engineering research community. I tried to google the topic, but cannot find any research paper addressing this topic. The authors also do not also cite any paper regarding this topic. (They cited the missing link prediction papers in other domains, but not for UAV swarms.)

Some motivational examples of missing link prediction (in the context of a UAV swarm network) should be given to highlight why it is important to address this problem.

There are also some problems with citations. For example, [2] and [3] are cited as missing link prediction methods. But I cannot find the keyword “missing link” in those papers.

Similarly, the “Common Neighbors” method is attributed to [3]. But, I cannot find this keyword in the paper.

Author Response

Dear reviewer:

Thank you in spite of being very busy to review my manuscript.Your comments are very helpful and thoughtful for me.According to your comments,there are following adjustments in my manuscript.Please see the attachment!

Best wishes to you!

2022.6.27

Round 2

Reviewer 2 Report

Since the cited paper by Wu Jun et al. containing information regarding missing link prediction for UAV swarms is in Chinese, the reviewer cannot verify it independently. The paper should be reviewed by an expert in the area of robotics/UAV to assess the merit of the proposed problem.

Author Response

Dear reviewer:

Thank you in spite of being very busy to review my manuscript.Your comments are very helpful and thoughtful for me.According to your comments,there are following adjustments in my manuscript.Please see the attachment!

Best wishes to you!

2022.7.1
